# Sorafenib and Doxorubicin Show Synergistic Effects in Human and Canine Osteosarcoma Cell Lines

**DOI:** 10.3390/ijms23169345

**Published:** 2022-08-19

**Authors:** Ya-Ting Yang, Vilma Yuzbasiyan-Gurkan

**Affiliations:** 1College of Veterinary Medicine, Michigan State University, East Lansing, MI 48824, USA; 2Department of Microbiology & Molecular Genetics, College of Veterinary Medicine, Michigan State University, East Lansing, MI 48824, USA; 3Department of Small Animal Clinical Sciences, College of Veterinary Medicine, Michigan State University, East Lansing, MI 48824, USA

**Keywords:** osteosarcoma, sorafenib, doxorubicin, combination therapy

## Abstract

Osteosarcoma (OSA) is the most common bone tumor in both humans and dogs and has a nearly ten-fold higher incidence in dogs than humans. Despite advances in the treatment of other cancers, the overall survival rates for OSA have stagnated for the past four decades. Therefore, there is a great need to identify novel and effective treatments. We screened a series of tyrosine kinase inhibitors and selected sorafenib, a multi-kinase inhibitor, for further evaluation alone and in combination with cisplatin, carboplatin, and doxorubicin on canine and human OSA cell lines. Our data point to synergistic effects when sorafenib is combined with doxorubicin, but not when combined with cisplatin or carboplatin, in both human and canine OSA. Based on current findings, clinical trials using a combination of doxorubicin and sorafenib in proof-of-concept studies in dogs are warranted. These studies can be carried out relatively quickly in dogs where case load is high and, in turn, provide useful data for the initiation of clinical trials in humans.

## 1. Introduction

Osteosarcoma (OSA) is the most common primary bone tumor of both humans and dogs with about 1000 OSA cases reported in humans and 10,000 cases in dogs every year in the United States [1]. Canine osteosarcoma is an aggressive cancer; around 80–90% of dogs present with micro-metastasis disease when diagnosed with osteosarcoma in clinics. Canine OSA (cOSA) closely resembles human OSA (hOSA), including in histopathological appearance, molecular markers, and response to conventional chemotherapies. As humans and dogs share similarities in their genome and living environments, dog disease is an excellent parallel disease with which to study osteosarcoma. The current first-line chemotherapies in cOSA are cisplatin, carboplatin, and doxorubicin [2,3,4]. With the current standard care, human patients show a high recurrence rate and develop resistance to chemotherapy. To overcome these clinical challenges, effective and well-tolerated therapeutic agents are needed for both humans and dogs. 

Many studies have evaluated the genomic complexity of OSA and reported somatic copy-number and epigenetic changes [5,6,7,8]. A genome-wide association study identified risk loci, including *CDKN2A/B*, *AKT2,* and *BCL2* [9]. Alterations in genes in the MAPK and PI3K/AKT pathways were found in 17% and 37% of cOSA tumor samples [6]. In addition, mutations in *SETD2* were found in 21% of examined tumor samples in three breeds tested (golden retriever, greyhound, and Rottweiler) [5]. Another study reported SETD2 mutations in 42% of cOSA tumors [6]. One study focused on epigenetic control via comparative DNA methylation in both hOSA and cOSA, the comparison between the two species identifying strong correlations between key transcriptional patterns [7]. The molecular similarities shared by hOSA and cOSA support the use of dogs as animal models. Findings from osteosarcoma sequencing have increased our understanding of the complexity of the tumor landscape. However, in both species, although the driver mutations are not yet been identified, several key pathways are highly activated in OSA. Activation of the P13K/AKT and MAPK pathway in OSA patients has been documented [6,10]. With the identification of key signaling pathways involved, more targeted therapeutic agents can be utilized and need to be evaluated. One of the potential candidates is sorafenib. 

Sorafenib (BAY 43-9906, Nexavar^®^, Whippany, NJ, USA), an oral small molecular inhibitor, inhibits a variety of signal transduction pathways. Sorafenib was primarily developed as a RAF inhibitor but was found to inhibit other targets, including VEGFR-2 (vascular endothelial growth factor receptor), VEGFR-3, PDGFR (platelet-derived growth factor receptor), RAF-1, B-RAF, and c-KIT [11,12,13]. Sorafenib is indicated for the treatment of patients with hepatocellular carcinoma (HCC) [14,15] as well as renal [16,17], hepatic [18], and thyroid [19] cancer in humans. In the literature, there are limited reports of using sorafenib to treat OSA. These studies include several clinical trials, two that used sorafenib as a single agent [20,21] and one that combined it with an mTOR inhibitor [22], as well as a case study that combined sorafenib with a RANKL inhibitor, denosumab [23]. In dogs, however, there have not been any clinical trials. There was one recent study that reported the tolerable dosage of sorafenib in a small group of dogs with various cancers [24], which showed that sorafenib was well tolerated, up to 3 mg/kg, given in three to eight doses. While sorafenib has been studied in other solid tumors, evaluation of the potential for sorafenib in treatment of cOSA and hOSA has been limited. So far, most studies on the anti-tumor activity of sorafenib were carried out with regard to HCC. Sorafenib alone has been used as a standard of care for HCC patients since 2007, and the combination of sorafenib and doxorubicin has been used in clinical trials to treat advanced (HCC) [14,15,25]. Doxorubicin, a topoisomerase II inhibitor, causes DNA damage by disrupting topoisomerase-II-mediated DNA repair and generation of free radicals and is one of the first-line chemotherapies for hOSA and cOSA [26,27]. However, there is very limited knowledge of using a combination of sorafenib and doxorubicin in OSA. 

In this study, we first screened a panel of ten tyrosine kinase inhibitors (TKIs) on three OSA cell lines and selected three TKIs (sorafenib, sunitinib, and gefitinib) for further studies. We identified sorafenib as the potential therapeutic candidate based on the IC_50_ values; the IC_50_ values of sunitinib and gefitinib were above the achievable plasma concentrations reported in the literature. The focus of this study was to evaluate the effects of a tyrosine kinase inhibitor, sorafenib, on cOSA and hOSA cells alone and in combination with current chemotherapeutic agents for OSA. We report here that sorafenib alone showed growth inhibition effects in both hOSA and cOSA cell lines. Furthermore, we show that sorafenib, alone and in combination with doxorubicin, had synergistic effects on cOSA and hOSA cells. These preclinical findings suggest that the multi-kinase inhibitor sorafenib should be considered for use in future clinical trials alone as well as in combination with doxorubicin and that dogs can serve as proof-of-concept studies for such trials. 

## 2. Results

### 2.1. Sorafenib, Gefitinib, and Sunitinib Showed Growth Inhibition Potential

As seen in Figure 1A, after 48 h of incubation, cell viability was measured across 10 tyrosine kinase inhibitors. Among these, only sorafenib, sunitinib, and gefitinib were capable of reducing cell viability to below 10% at 100 μM. Thus, we selected these three TKIs for further studies. 

### 2.2. Cytotoxicity Assay with Sorafenib and Conventional Chemotherapeutics for OSA Treatment 

Three TKIs, gefitinib, sorafenib, and sunitinib, were selected for further evaluation. The effect of sorafenib and other drugs on cell growth inhibition of four cOSA cell lines (D17, Abrams, BZ, and Gracie) and three hOSA cell lines (SAOS2, U2OS, and MG63) was examined via MTS assay. After 72 h of incubation, dose–response curves were generated based on cell viability at a range of concentrations, and IC_50_ values were calculated using GraphPad Prism software (Figure 1B,C). Two TKIs, gefitinib [28] and sunitinib [29], which displayed IC_50_ values above the achievable plasma concentrations reported in the literature, were not included in further studies.

The IC_50_ values for the conventional chemotherapeutics cisplatin, carboplatin, and doxorubicin were also determined in our cell lines (Table 1). Cisplatin [30] and carboplatin [31] were found to have IC_50_ values larger than the reported achievable plasma concentration in the literature. However, the IC_50_ values for doxorubicin ranged from 58–226 nM in our cell lines, which were lower than a documented achievable plasma concentration of 1130 nM in dogs. Additionally, the IC_50_ values for sorafenib were from 3–9 μM, whereas studies reported achievable plasma concentrations of 13 μM in humans and 6.45 μM in dogs. These comparisons with reported achievable plasma concentrations provided the rationale to include sorafenib and doxorubicin for further evaluation, in order to select combinations of drugs each with the potential to reach therapeutic effects within clinically relevant doses [32,33]. 

### 2.3. Cell Migration Ability Was Inhibited by Sorafenib

As shown in Figure 2, the effect of sorafenib on the migration capacities of OSA cells was determined by a wound-healing assay. D17, Abrams, and SAOS2 cells were treated with or without sorafenib for up to 48 h of incubation time after performing a scratch at the center of a 6-well plate. In this assay, sorafenib effectively suppressed the migration of SAOS2 and Abrams cell lines. To avoid cell cytotoxic effects, we used a dose of sorafenib lower than the IC50s in this study. After 24- and 48-hour treatment, SAOS2 cells incubated with sorafenib migrated less than the control group (Figure 2). 

### 2.4. Sorafenib Inhibited STAT3 and ERK Phosphorylation in the Abrams and D17 Cell Lines

Two major signal transduction pathways activated by receptor tyrosine kinases are the PI3K/AKT (phosphatidylinositol 3-kinase/serine-threonine protein kinase) and the MAPK/ERK (mitogen- activated protein kinases/extracellular signal-regulated kinases) pathways [37,38]. In Figure 3, the potential mechanisms of sorafenib treatments were investigated using Western blot analysis of key pathways. In the canine Abrams cell line, p-ERK and p-STAT3 decreased at 10 μM or higher concentrations of sorafenib. The D17 cell line also displayed complete loss of p-ERK upon treatment with sorafenib at 10 μM, while in SAOS2 this pathway appeared unaffected by the treatment.

### 2.5. Sorafenib and Doxorubicin Showed Synergistic Effects

We next assessed the possibility of improving anti-tumor effects through combinations of drugs with different modes of action. We examined three drugs (cisplatin, carboplatin, and doxorubicin) in combination with sorafenib. Two OSA cell lines (D17 and SAOS2) were exposed to various concentration of sorafenib, one of the other three drugs, or their combination for 72 h. Later, cell viability was examined using MTS assay and the effects were calculated using CompuSyn software (Combosyn, Paramus, NJ). As seen in Figure 4A, sorafenib combined with doxorubicin resulted in several combination index values of less than 1, indicating synergistic effects. Most of the combinations of cisplatin (Figure 4B) and carboplatin (Figure 4C) showed antagonistic effects with sorafenib in the D17 and SAOS2 cell lines.

### 2.6. Sorafenib and Doxorubicin Induced Cell Cycle Arrest

We studied the effects of cell cycle changes with sorafenib and doxorubicin use on three OSA cell lines and found cell arrest at G2/M phase in all cell lines with their combined use. The distributions are very similar after treatment with 5 μM sorafenib alone (Figure 5A,B), whereas G2/M arrest was clearly augmented when 5 μM sorafenib and 100 nM doxorubicin were used together (Figure 5D), changing G2/M from 10% to 91% and resulting in a cell arrest at the G2/M phase. This observation was consistent among all three OSA cell lines (D17 (Figure 5E), Abrams (Figure 5F), and SAOS2 (Figure 5G)). Interestingly, the apoptotic fraction induced by doxorubicin alone was reduced in the presence of sorafenib (Figure 5C,D).

## 3. Discussion

In bone cancers, targeting the MAPK/ERK pathway as a therapeutic strategy has been studied in vitro [12,39,40], in vivo [41,42,43], and in clinical trials [22,44]. The inhibitory effects of sorafenib on the MEK/ERK signaling pathway have been documented [12,39,44]. In one study, around 67% (20/30 samples) of OSA tumors in human patients demonstrated immunopositivity in histopathology for p-ERK, indicating that the ERK pathway is highly activated in hOSA and that blocking p-ERK may be a potential therapeutic option [45]. STAT3 activation also plays a critical role in OSA cells as it supports cells survival and proliferation [45,46,47]. A previous study demonstrated that STAT3 activation contributes to the survival and proliferation of human and canine OSA cells, thereby suggesting that STAT3 is a potential target for therapeutic strategy [46]. Previous studies reported that sorafenib as well as its derivative sc-1 were capable of downregulating the activation of STAT3 and ERK pathways and reducing tumor volumes [47]. Our study on the canine Abrams OSA cell line shows similar results in that sorafenib downregulated the expression of p-STAT3 and p-ERK. Besides the MAPK/ERK and STAT3 signaling pathways, another target for sorafenib is VEGFA. A recent study investigated somatic copy-number alternations (SCNA) as an approach to identify potential targeted drugs to treat different amplifications in an osteosarcoma patient-derived xenograft model (PDTX) [8]. In the VEGFA-amplified PDTX model of OSA, sorafenib resulted in reduced tumor volume compared to vehicle, whereas in another PDTX model without VEGFA-amplified OSA, there was no benefit from sorafenib treatment. This finding suggests that sorafenib could be used in OSA patients with VEGFA-amplified tumors as well. The observation of induction of apoptosis by sorafenib is consistent with a previous study where an increase of caspase-3 activity in sorafenib-treated cells was documented in the D17 cell line [48].

In the cell cycle assay, we compared three different concentrations (1.25, 2.5, and 5 μM) of sorafenib. Sorafenib alone from 1.25 μM to 5 μM did not have an effect on the phases of the cell cycle, which is consistent with a previous study [48]. In addition, cisplatin demonstrated an antagonistic effect with sorafenib (Figure 4C), in agreement with previous findings of combining sorafenib and carboplatin (a derivative of cisplatin) in the D17 cell line [48]. Although sorafenib and doxorubicin as single treatments have been studied extensively, the present study is the first to show the further efficacy of combining sorafenib and doxorubicin in inhibiting cOSA cell growth. 

In humans, sorafenib has been used in a few clinical trials for osteosarcoma. Two clinical trials in osteosarcoma patients were reported in recent years. A phase II non-randomized trial explored sorafenib treatment in 35 patients with relapsed and unresectable OSA and reported that the median survival was 7 months [21]. In another non-randomized phase II clinical trial, researchers used sorafenib and everolimus to treat a group of 38 high-grade osteosarcoma patients. Among these patients, a subgroup of patients with overexpression of both P-ERK1/2 and P-RPS6 responded better than double negative patients, where the progression-free survival was 7 months and 2 months, respectively [22]. These small cohorts indicate that using sorafenib alone or with other drugs to treat OSA has potential to improve outcomes, yet more robust studies are needed to determine a more personalized approach to identify the optimal drug for each patient. One limitation in these clinical trials is that the recruited patients all had advanced and relapsed osteosarcoma. There is, therefore, a lack of knowledge regarding using sorafenib alone or in combination with other chemo-therapeutic agents to treat early-stage patients. For sorafenib, 400 mg twice a day was identified as the maximum-tolerated dose and was used in various clinical studies on solid tumors [13,49]. The most adverse effects identified in these reports included lymphopenia and hypophosphatemia (16%), hand and foot syndrome (13–39%), fatigue (5–39%), oral mucositis (5–20%), diarrhea (5–55%), anemia thrombocytopenia (11%), and rash (19–61%) [16,22,49,50,51,52]. These toxic effects led to dose reductions, short interruptions, or discontinuation for these patients. One strategy to reduce these toxic effects is to use combination therapies. Therefore, involving sorafenib and other therapeutic agents such as doxorubicin can be more effective and may avoid intolerable adverse effects. In addition, combination chemotherapies with drugs utilizing different mechanisms of action have the potential to decrease the possibility of drug resistance and reduce drug dosage while maintaining treatment efficiency. Interesting approaches such as the use of selenium nanoparticles loaded with sorafenib have been recently described [53].

In dogs, there are very limited reports on sorafenib use. One report published pharmacokinetic studies on sorafenib in healthy dogs, and showed that 60 mg/kg per day was well tolerated in a 4-week study [35]. To date, there are only a few studies of tolerance in a small group of dogs with different types of cancer [24,54,55,56]. The first report of sorafenib tolerance in 12 client-owned dogs documented that these dogs tolerated the drug up to a dose of 3 mg/kg for 3 to 8 weeks of treatment. In a case study of canine transitional carcinoma, oral sorafenib doses initiated at 4 mg/kg/day increased to 10 mg/kg/day. The dog remained with stable disease for three months [54]. Among these four studies, only one addressed the efficacy of sorafenib. A total of 13 dogs with unresectable HCC were separated into two groups: sorafenib (5 mg/kg twice daily) and metronomic chemotherapy (thalidomide, piroxicam, and cyclophosphamide). The overall survival was 361 days in dogs receiving sorafenib compared to 32 days in dogs with metronomic chemotherapy [56]. A recent study included six dogs with different types of tumor burden that were given sorafenib, 3 mg/kg, to determine tolerance and peak plasma concentrations (Cmax). All dogs tolerated the dose and showed no adverse events. The average (Cmax) was 118.2 ± 72.1 nM with a median time to peak of 4 h (range 2–12 h) post dosing [55], showing a fairly large variability in pharmacokinetics in dogs. These studies provide a basis to initiate future clinical trials in cOSA to evaluate therapeutic efficacy. Additionally, in human clinical trials, the recruited patients for new drug studies are those who have failed to respond or relapsed after first-line treatments [21,22]. However, in canine patients, studies can be initiated in both naïve and relapsed cases, and such information can facilitate human clinical trials. 

The use of other TKIs such as dasatinib [57,58] and toceranib [59,60,61,62] have been reported in small numbers of osteosarcoma patients for oral tolerability and safety verification in the canine oncology field. Two clinical studies reported the potential of dasatinib for treating cOSA. One of studies was a drug screening containing 86 small molecule kinase inhibitors on cells derived from one canine patient and reported that the IC_50_ values for dasatinib and sorafenib were 0.15 and 9.5 μM, respectively, suggesting that dasatinib was the most promising therapeutic agent. The patient survived for more than 730 days after the initial diagnosis after amputation and five cycles of carboplatin, followed by dasatinib treatments [57]. Another study reported four cOSA patients with reported survival times ranging from 456–1003 days with treatment using dasatinib at 0.5–0.75 mg/kg every day or every other day for 6.5 to 25 months and that one dog was still alive at the time of publication of the paper, at which time it was 1003 days post treatment [58]. These two reports indicate potential clinical benefits of dasatinib as an adjuvant treatment for cOSA, but larger-scale studies are needed. Toceranib phosphate, which was the first TKI approved for veterinary use, is the most-utilized molecularly targeted agent in the United States. Toceranib targets receptor tyrosine kinases, including C-KIT, VEGFR-2, PDGFa/b, and CSF-1 (colony stimulating factor–1) and is used as a first-line treatment for dogs with mast cell tumors [63]. The first clinical study to use toceranib in cOSA reported that it contributed to clinical benefit on 11 out of a total of 23 cOSA patients, as these 11 patients presented with either partial response or stable disease [64]. Later on, two small cohorts containing 20 [59] and 22 [60] cOSA patients with lung metastasis reported a different conclusion, as canine patients treated with amputation and adjuvant chemotherapies and toceranib had median survival times of 90 and 89 days, respectively, vs. 76 [65] and 95 [66] days in reported studies of metastatic cOSA patients, indicating limited clinical benefit. While those two studies showed that using toceranib did not have a therapeutic effect on canine patients with metastatic osteosarcoma [59,60], other studies explored the combination of toceranib with other therapeutic agents in cOSA without metastatic disease. A clinical study that reported on 10 cOSA patients treated with toceranib and carboplatin resulted in overall survival times of 253 days [61], which was lower than the previous published median survival of 321 days from a study of 48 cOSA patients treated with amputation and four cycles of carboplatin [67]. Another study of 126 cOSA patients compared the outcomes of adding toceranib to treatments with carboplatin/piroxicam/cyclophosphamide [62]. In the treatment group with the addition of toceranib, the overall survival time was 318 days, while the overall survival time in the control group was 242 days; however, statistical analysis found no clinical benefit from toceranib [62]. These findings are not surprising, since toceranib mainly targets c-KIT, and c-KIT is not a major driver in OSA. In addition, the effects of toceranib on other targets such as VEGFR-2, PDGFa/b, and CSF-1 are not fully studied in canine OSA. One clinical trial with 10 cOSA patients reported that VEGF levels did not change over time in cOSA patients with toceranib treatments [61]. Additionally, a recent study documented that toceranib treatments did not change the expression of VEGFR-2, PDGFa/b, and c-KIT from the control group in an in vivo study [68]. The expression of VEGFR-2 varied in different OSA cell lines and tumors [8,69], which suggests the potential of using toceranib in patients with overexpressed VEGFR-2, as well as the significance of individual targeted therapy. Sorafenib and dasatinib, on the other hand, inhibit multiple receptor tyrosine kinases that are relevant in OSA. The data from our study also support the use of sorafenib alone and in combination with doxorubicin in canine clinical trials. 

The combination of sorafenib and doxorubicin has been studied and applied in clinical trials on hepatocellular carcinoma (HCC) [25,70,71]. An earlier clinical trial in 2010 reported that advanced HCC patients receiving sorafenib and doxorubicin together had a longer survival time compared to doxorubicin alone [25]. However, two more recent clinical trials addressed the question of using sorafenib and doxorubicin. A phase III clinical trial reported no differences in overall or progression-free survival between HCC patients treated with sorafenib alone or with sorafenib plus doxorubicin [70]. In another clinical trial, doxorubicin was added after HCC patients showed radiologic signs of progression on sorafenib, but there was no improved outcome when compared to historical controls [71]. It is likely that molecular heterogeneity among the HCC patients contributed to the different results; thus, including molecular characterization of tumors in future clinical trials would help delineate those patients that may benefit from each treatment.

In this present study, we demonstrated that sorafenib, a multi-kinase inhibitor, acts as an effective drug against both human and canine osteosarcoma cells. The IC_50_s on OSA cell lines ranged from 3–9 μM, which is considerably lower than that of a human fibroblast cell line, MRC-5, 19.7 μM, as indicated by a recent report [72]. We provided evidence of anti-proliferation and migration inhibition effects in vitro. In the canine Abrams cell line, the activation of ERK and STAT3 pathways was inhibited by sorafenib at 10 μM. However, in the PI3K-AKT pathway, p-AKT remined unchanged with the treatments, even at the highest dose. As we explored the potential of drug combinations, sorafenib and doxorubicin demonstrated a synergistic effect and resulted in cell arrest at the G2/M phase in D17 and Abrams. These findings indicate that sorafenib, alone and in combination with doxorubicin, can be used for novel therapeutic strategies for the treatment of certain canine and human OSAs that show activation of the ERK pathway. 

## 4. Materials and Methods

### 4.1. Cell Culture

Canine OSA cell line D17 and human OSA cell line (SAOS2, U2OS, and MG63) were purchased from ATCC. The Abrams canine OSA cell line was provided by Dr. Elizabeth McNeil and originally established and shared by Dr. Doug Thamm. The BZ canine OSA cell line was established by our laboratory from a German shepherd dog. D17 was derived from a lung metastasis of a poodle; the canine Abrams cell line was derived from a metastatic nodule and the canine Gracie cell line was derived from a primary tumor sample. As indicated in the ATCC repository, the SAOS2 cell line was derived from a primary tumor in a Caucasian female, U2OS was from a tibia biopsy sample from a Caucasian female, and MG63 from a Caucasian male. 

For cell culture maintenance, human OSA cell lines were incubated with DMEM medium and canine OSA cell lines were maintained with a-MEM medium, and all cells were supplemented with 10% fetal bovine serum and incubated in a humidified incubator at 37 °C with 5% CO_2_. 

### 4.2. Compounds Used in Drug Screening

Sorafenib was purchased from LC laboratories (Woburn, MA, USA). Sorafenib was dissolved in DMSO while cisplatin was dissolved in PBS, and carboplatin was dissolved in water. Other TKIs, including cladribine, dasatinib, erlotinib, gefitinib, masitinib, nilotinib, sorafenib, sunitinib, toceranib, and tozasertib were all purchased from Sigma-Aldrich and dissolved in DMSO.

### 4.3. Cell Viability Assay (MTS Assay) and Small Panel Drug Screening with TKIs

The MTS assay (Promega Corp., Madison, MI, USA) was used to determine the IC_50_ values of sorafenib and other compounds on OSA cells. Cells were seeded in 96-well plates at a density of 2500–3500/well. After 24 h, cell culture medium was replaced by complete medium with each compound at the designated concentrations. Cells were treated for 72 h. Cell viability was analyzed using CellTiter 96 Aqueous Non-Radioactive Cell Proliferation Assay (MTS) and determined by the amount of colored formazan dye produced by live cells. The absorbance of the formazan dye produced was measured at wavelength 490 nm, and IC_50_ values were calculated using PRISM Statistical Software. Each concentration for each drug was assayed in triplicate for the IC_50_ calculations.

In this study, we treated three OSA cell lines (D17, Abrams, and SAOS2) with a small panel of compounds including ten tyrosine kinase inhibitors (cladribine, dasatinib, erlotinib, gefitinib, masitinib, nilotinib, sorafenib, sunitinib, toceranib, and tozasertib) for 48 h. Percent growth inhibition was calculated for each treatment with the MTS assay with the vehicle control treatment containing 1% DMSO. For all TKIs, we used a high drug dose of 100 μM.

### 4.4. Wound Healing Assay

The wound healing assay was used to examine the migration capacity of cells in a monolayer. Briefly, 100,000 cells/well were plated overnight and allowed to reach 70–80% confluence in 6-well plates. Then, scrapes were made on the plates using a 1-mL pipette tip. The cells were then incubated with cell culture medium with or without sorafenib (D17: 3 μM, Abrams: 4 μM, SAOS2: 3 μM). Each scrape was photographed after being made and at each specific time point thereafter.

### 4.5. Combination Index (CI)

Cells were simultaneously incubated with two compounds at a fixed ratio (sorafenib: doxorubicin = 20:1, 50:1, or 100:1; sorafenib: cisplatin = 4:1; and sorafenib:carboplatin = 1:5) for 72 h. The synergistic effects of each pair of drugs were determined via isobologram and combination index (CI) analysis using CompuSyn software (Combosyn, Paramus, NJ, USA). The analysis was adapted from the median-principle methods of Chou and Talalay [73]: results of CI < 1, CI = 1, and CI > 1 indicate synergism, addition, and antagonism, respectively.

### 4.6. Cell Cycle Analysis

Cells were treated with 1% DMSO (control), sorafenib (1.25, 2.5, or 5 μM), doxorubicin (25, 50, or 100 nM), or combination (ration 50:1) for 24 h, then collected by centrifugation and fixed with 70% ethanol at 4 °C overnight. The ethanol was removed via centrifugation, and cellular DNA was stained with propidium iodide (50 μg/mL) containing RNase (1 mg/mL). After cells were stained for at least 4 h, the PI fluorescence of individual nuclei were recorded with FACScan. The quantitative assessment of cell cycle phase and apoptosis were then determined using Modfit LtTM software after correction for debris and aggregate cell populations.

### 4.7. Protein Analysis Using Western Blots

The Abrams OSA cells (500,000/well in 6-well plate) were treated with either vehicle (0.1% DMSO) or sorafenib for 24 h. Cells were lysed with 250 μL of CelLytic M lysis buffer (C2978, Sigma-Aldrich, St. Louis, MO, USA), 2 μL of protease inhibitor (P8340, Sigma-Aldrich), and 2 μL of phosphatase cocktail inhibitor B (sc-45045, Santa Cruz, Dallas, TX, USA) according to manufacturer protocol. Protein concentrations were quantified with the QubitTM Protein Assay Kit. A total of 60 μg of protein per well was loaded on Bolt Bis-Tris 4–12% polyacrylamide gels (Thermo Fisher Scientific Inc.) and transferred to polyvinylidene difluoride membranes. The membranes were incubated with 5% bovine serum albumin (BSA) for 2 h at room temperature, then incubated with the following primary antibodies at 4 °C overnight to detect antigen: ERK (1:500), p-ERK (1:250), STAT3 (1:500), p-STAT3 (1:500), β-tubulin (1:4000) (Cell Signaling Technology). After three washes in tris-buffered saline with 0.05% Tween 20, the membranes were incubated with appropriate secondary antibody (donkey anti-mouse (1:15,000) or goat anti-rabbit (1:15,000)) for 1 h at room temperature. The membranes were visualized using the Odyssey Infrared Imaging System (LI-COR Biosciences, Lincoln, NE, USA) and analyzed using Image Studio™ Lite software (LI-COR, Lincoln, NE, USA).

### 4.8. Statistical Analysis

Our results represent at least three separate experiments. Statistical analysis was performed with GraphPad Prism (8.0.0, Graph Pad Software Inc., San Diego, CA, USA) and differences between categories were analyzed with one-way ANOVA; *p*-values of <0.05 were considered to be significant.

## 5. Conclusions

In conclusion, the findings from our studies suggest that the tyrosine kinase inhibitor sorafenib exhibits anti-tumor activity on its own through the inhibition of proliferation and the induction of apoptosis via inhibition of the ERK/MAPK and STAT3 pathways in OSA cells. Sorafenib is a comparatively novel drug in osteosarcoma; therefore, information on large-scale clinical data and other drug combinations is limited. In addition, combining sorafenib with doxorubicin was found to be synergistic in clinically relevant doses, inducing cell cycle arrest in the G2/M phase. Based on current findings, clinical trials using the combination of doxorubicin and sorafenib in proof-of-concept studies in dogs are warranted. These studies can be carried out relatively quickly in dogs where case load is high and, in turn, provide useful information for the initiation of clinical trials in humans.

## Figures and Tables

**Figure 1 ijms-23-09345-f001:**
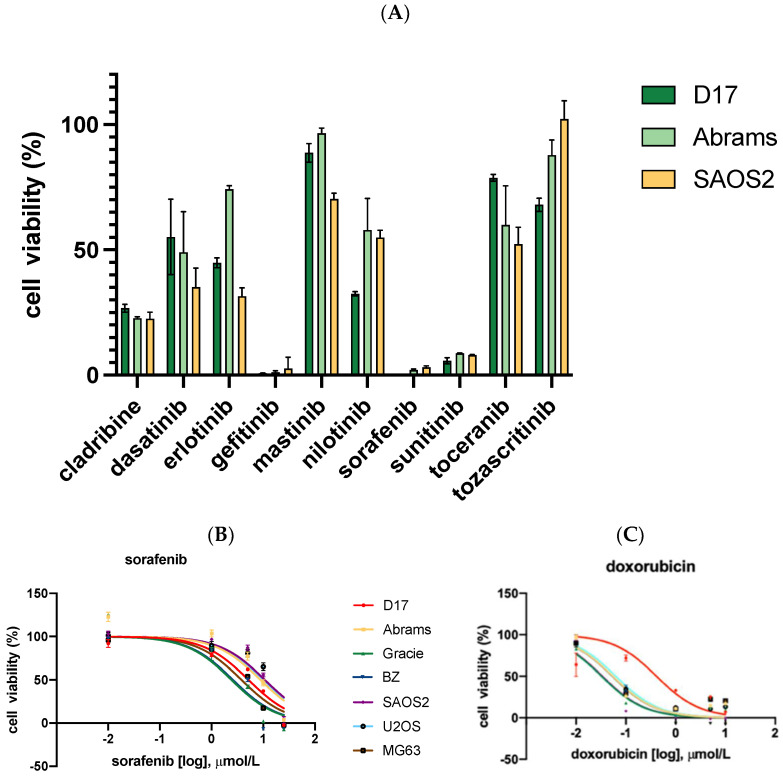
Cell viability was measured by CellTiter-Glo assay on all seven OSA cell lines, including four canine OSA (D17, Abrams, Gracie, and BZ) and three human OSA (SAOS2, U2OS, and MG63) cell lines. All cell lines were treated with drugs for 72 h. (**A**) Ten TKIs demonstrate different levels of inhibition on OSA cell viability. (**B**) OSA cell lines were treated with sorafenib for 72 h. (**C**) OSA cell lines were treated with doxorubicin.

**Figure 2 ijms-23-09345-f002:**
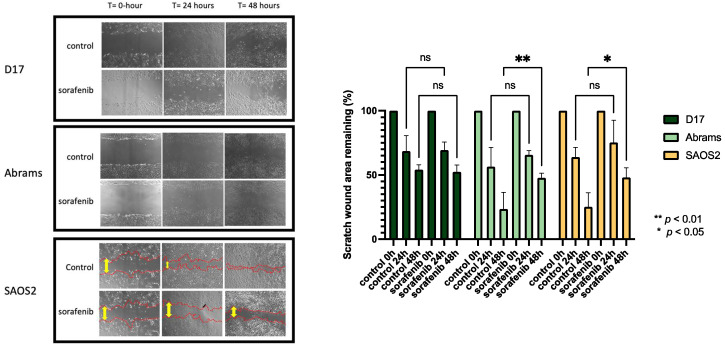
Photomicrographs taken with Nikon camera at 40× magnification, comparing wound healing in three OSA cell lines with and without sorafenib for up to 48 h. (D17: 3 μM, Abrams: 4 μM, SAOS2: 3 μM). **: *p* < 0.01, *: *p* < 0.05, and ns (not significant) for sorafenib treatment compared to control as determined by two-way ANOVA with Tukey’s multiple comparison test.

**Figure 3 ijms-23-09345-f003:**
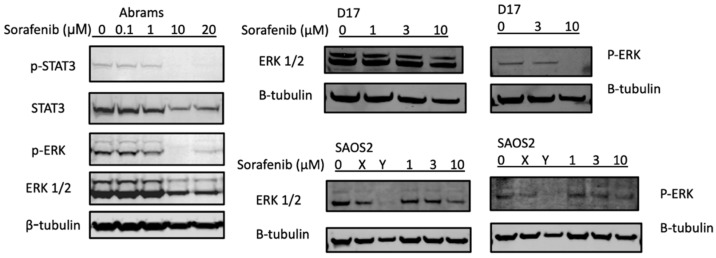
Sorafenib decreases expression of p-STAT3 and p-ERK in protein analysis. Canine osteosarcoma Abrams cells were treated with either DMSO (control) or various concentrations of sorafenib (0.1, 1, 10, 20 μM) for 24 h then subjected to Western blot analysis. β-actin and β-tubulin were used as loading controls. X: JQ1 0.1 μM, Y: bortezomib 25 nM.

**Figure 4 ijms-23-09345-f004:**
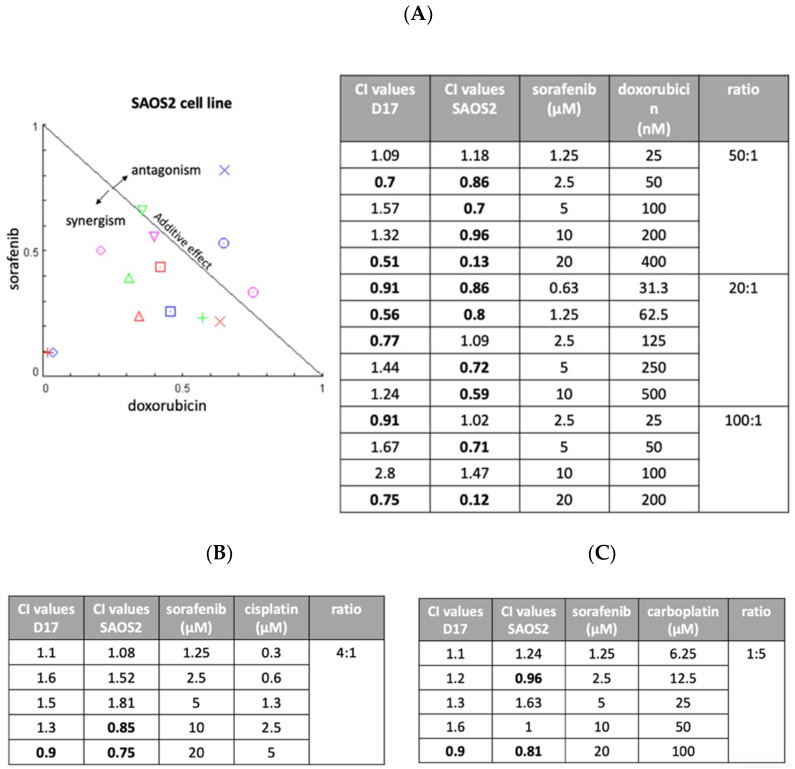
Three different ratios of sorafenib and doxorubicin were examined with combination index (CI) assay on SAOS2 and D17 cells. (**A**) Normalized isobologram. We included three different ratios of sorafenib and doxorubicin (20:1, 50:1, and 100:1). (**B**) The combination of sorafenib and cisplatin was examined on D17 and SAOS2 cells at a ratio of 4:1. (**C**) The combination of sorafenib and carboplatin was examined on D17 and SAOS2 cells at a ratio of 1:5.

**Figure 5 ijms-23-09345-f005:**
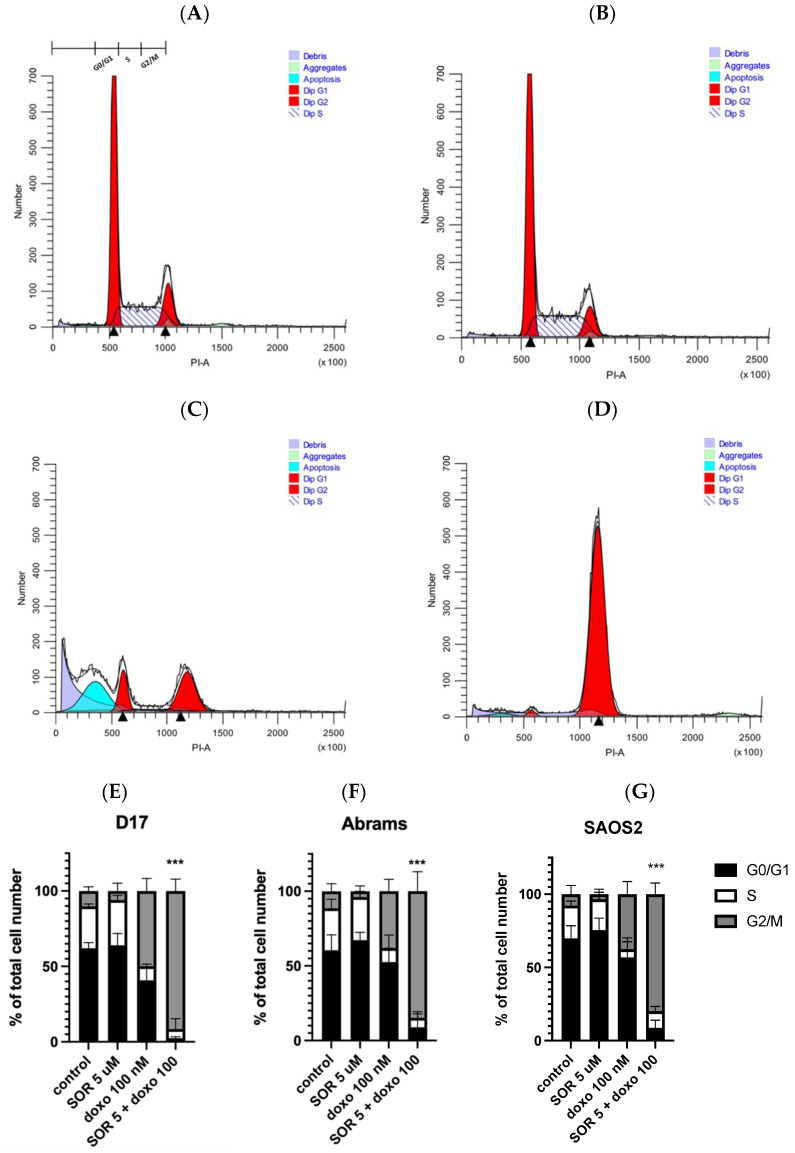
The combination of sorafenib and doxorubicin caused inhibition of cell cycle progression, resulting in G2/M arrest in D17 cells. Cell cycle distribution of D17 OSA cells treated with either (**A**) DMSO (control), (**B**) sorafenib 5 µM, (**C**) doxorubicin 100 nM, or (**D**) both sorafenib 5 µM plus doxorubicin 100 nM for 24 h. Representative flow histograms demonstrate changes in the cell cycle progression on canine OSA D17 cell line. The combination of sorafenib and doxorubicin resulted in a cell arrest at the G2/M phase. Representative cell cycle distribution graphs show G2/M cell arrest in (**E**) D17, (**F**) Abrams and human OSA, and (**G**) SAOS2 cell lines. ***: *p* < 0.001 for G2/M arrest compared to the combined treatment (5 µM sorafenib and 100 nM doxorubicin) as determined by one-way ANOVA with Dunnett’s multiple comparison test.

**Table 1 ijms-23-09345-t001:** The effects of sorafenib on cell viability and IC_50_s for sorafenib and three first-line OSA chemotherapy agents (cisplatin, carboplatin, and doxorubicin).

Drug IC_50_	Cisplatin (μM)	Carboplatin (μM)	Doxorubicin (μM)	Sorafenib (μM)
D17	4	45	0.2	6
Abrams	12	263	0.06	9
Gracie	2	41	0.06	4
BZ	12	273	0.05	3
SAOS2	3	47	0.07	7
U2OS	7	57	0.06	5
MG63	5	70	0.05	4
Reported maximum plasma concentration	4 μM (dog)	72 μM (dog)	1.13 μM (dog)	13 μM (human) 6.45 μM (dog) *
[reference]	[30]	[31]	[34]	[35,36]

*: A single intraduodenal sorafenib dose (up to 60 mg/kg) was given to anesthetized dogs in a PK study.

## Data Availability

The body of the manuscript contains all of the data.

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
