# Peer review of "Sorafenib and Doxorubicin Show Synergistic Effects in Human and Canine Osteosarcoma Cell Lines"

_ijms, 2022, doi:10.3390/ijms23169345_

Round 1
Reviewer 1 Report
The article is interesting and executed at a good methodological level.
A scheme of experiments is required.
The number of repetitions should be reflected under each figure. The reliability of the results should also be signed under each figure. As well as the statistical method. Fig. 1a. Spreads in bars (standard error of measurement) should be reported. Statistical differences should be reflected in the figure. Figure 2. The data should be calculated and presented as columns. Statistical data processing should be carried out. For Figure 3, the results should also be calculated statistically and presented as columns. Perhaps the authors will be interested in the article in their further research. Comparative Analysis of the Cytotoxic Effect of a Complex of Selenium Nanoparticles Doped with Sorafenib, "Naked" Selenium Nanoparticles, and Sorafenib on Human Hepatocyte Carcinoma HepG2 Cells - PubMed (nih.gov) The conclusion must be without reference. Should be written clearly and reflect the results of the work.
Author Response
Thank you for your thoughtful comments. We have addressed all of the points raised and the manuscript has benefited from your input.
Please find our point-by-point responses in the document attached.

Reviewer 2 Report
Dear authors.
Malignancies including osteosarcomas are still a serious and unresolved clinical issue. The search for new therapies, including combination therapy, is an important research topic.
1 In Figure 1A, the standard deviation should be supplemented.
2. in section 4.1, please add a short description to each cell line (one sentence).
3. Was an assessment of the repeatability of the width of the wound made in the migration test. Accurate execution with a pipette tip is very difficult and often not very repeatable. At what dose were the drugs tested - to be completed in the methodology.
4. Please supplement the paper with your results or by citation regarding the cytotoxicity of the drugs tested against normal fibroblast and osteoblast cells. The absence of such results significantly affects the assessment of the paper.
Author Response

(The authors gave the same response as above.)

Round 2
Reviewer 1 Report
The article has been substantially revised and is ready for publication. I wish the authors success in their future research.
Reviewer 2 Report
Dear Authors, I congratulate you on the work you have put into the preparation of the manuscript and thank you for addressing all the comments and making the appropriate changes to the manuscript.
I recommend the manuscript for publication in its current form.